# Fatigue Performance of Rib Beam Bridge Slabs Reinforced with Polyurethane Concrete Based on the Damage Theory

Yifan Wang [1], Tianlai Yu [1,*], Linlin Zhang [1,2], Lihui Yin [1,3], Yuxuan Wu [1] and Binglin Chen [1]

1   School of Civil Engineering, Northeast Forestry University, Harbin 150040, China;
    wangyifan2019@nefu.edu.cn (Y.W.); 21115013@bjtu.edu.cn (L.Z.); yinlihui@nefu.edu.cn (L.Y.);
    2021021099@nefu.edu.cn (Y.W.); binglin@nefu.edu.cn (B.C.)
2   School of Civil Engineering, Beijing Jiaotong University, Beijing 100044, China
3   School of Civil Engineering, Heilongjiang University, Harbin 150080, China
*   Correspondence: 2019115437@nefu.edu.cn

**Abstract:** In this paper, the rib beam bridge slabs were taken as the research object. Static load and fatigue tests were carried out on the benchmark bridge slabs to determine the ultimate load capacity and fatigue life of the bridge slabs. Then, the bridge slab was pre-damaged and reinforced with polyurethane concrete. A fatigue test was carried out on the reinforced bridge slab to study the fatigue performance. Based on the damage theory, the fatigue damage reinforcement finite element models of the bridge slabs under different damage degrees were established. The fatigue performance of the reinforced bridge slabs was systematically studied. The results show that the fatigue damage of the reinforced bridge slab developed in stages. Compared to the unreinforced bridge slab, the fatigue damage of the reinforced bridge slab was significantly reduced at each stage. According to the least square method and numerical analysis results, a residual-bearing-capacity model including damage degree and fatigue cycles of the reinforced bridge slabs is proposed, which can be used as a reference in bridge slab reinforcement design.

**Keywords:** polyurethane concrete; bridge slab; fatigue test; reinforcement; damage theory





## 1. Introduction

Medium- and small-span bridges are widely used in highway construction with their excellent economy and applicability in China. The proportion of medium- and small-span bridges in highway bridges in China reached 86.15% in 2020 [1]. Among them, the rib beam bridge occupies a large proportion due to its low-structural weight, short construction period, and good economic performance [2]. As a crucial part of the rib beam bridge, the bridge slab directly carries the wheel load of passing vehicles. Under the long-term heavy load and overload of vehicles, the bridge slabs often cause concrete spalling and cracking as well as steel corrosion due to fatigue damage, which further reduces the ultimate load capacity of the bridge and affects the service life of the bridge.

To explore the influence of fatigue damage on the mechanical properties of the bridge slabs, Huang [3] carried out fatigue tests on T-beams with different corrosion degrees. The results showed that the fatigue life of T-beams corroded by steel bars decreased significantly under the fatigue load. Under certain fatigue load levels, when the corrosion rates of steel bars were 1.5% and 3.43%, the fatigue life of the test beams decreased by 65.84% and 75.40%, respectively. Liu [4] carried out fatigue tests on three concrete rectangular beams under different load levels. The results showed that the stiffness of the concrete beams decreased by 62.7%, 53.7%, and 44.7% under the fatigue load levels of 0.6, 0.7, and 0.8, respectively. Li [5] carried out fatigue tests on T-beams with different length ratios. The results showed that the smaller the length ratio, the lower the fatigue damage of the bridge deck. When the length ratio was 1.66, the residual ultimate load capacity of the bridge deck decreased by 56.3%. In summary, fatigue damage has a great influence on the safety and durability of

bridges, and can even lead to serious accidents. Therefore, it is particularly important to reinforce and repair a bridge deck with fatigue damage.

At present, the bridge slab reinforcement methods can be mainly divided into the following: increase section method, bonding steel plate method, and bonding fiber materials; although the above methods can repair the fatigue damage of bridge slabs to a certain extent, the effect of the traditional reinforcement method is not ideal due to the long construction period and low bonding strength between the reinforcement materials and bridge deck. With its excellent mechanical properties, corrosion resistance, and good bonding performance with stone, steel, and other inorganic non-metallic materials, polyurethane concrete has gradually been applied to bridge structure reinforcement

Liu [6] carried out static load tests on hollow slab beams under different reinforcement methods. The results showed that the ultimate load capacity of the bridge deck reinforced by polyurethane concrete was 71.2% higher than that of the original beam and 483.2% higher than that of the bridge deck reinforced by the increase section method. Hussain Haleem K [7] strengthened the pre-damaged T-beam with polyurethane concrete. The results showed that the ultimate load capacity of the strengthened T-beam was 170% higher than that of the original beam, and the crack width was reduced by 58%. Gao [8] carried out polyurethane concrete reinforcement on the tensile zone of a concrete hollow slab beam and explored its reinforcement effect through a static load test. The results showed that the interface bonding performance between the polyurethane concrete reinforcement layer and concrete was good. The ultimate load capacity of the bridge slab plate after reinforcement was 41.67% higher than that of the original beam and 25.92% higher than that of the concrete reinforcement.

At present, most of the research on bridge slabs reinforced with polyurethane concrete is around the static ultimate load capacity of the bridge slabs, and the fatigue performance of the bridge slabs after reinforcement is relatively less, while the fatigue study of the damaged bridge slabs after reinforcement is almost non-existent. Therefore, this paper used the rib beam bridge slabs as the research object, conducted a pre-damage fatigue test on the bridge slabs, and carried out a fatigue test on the reinforced polyurethane concrete bridge slabs after damage. Based on the damage theory, the numerical simulation of the bridge slabs under different damage degrees was carried out. According to the least square method and the numerical analysis results, a residual-bearing-capacity model including damage degree and fatigue cycles of the reinforced bridge slabs is proposed, which can be used as a reference in bridge slab reinforcement design.

## 2. Materials and Methods

### 2.1. Test Overview

2.1.1. Design of Bridge Slabs

Bridge Slabs Size Design

In this experiment, three rib beam bridge slab models were designed. Among them, two were reference bridge slab models (beams P1 and P2): P1 was subjected to static load tests to determine the ultimate load capacity, and P2 underwent fatigue tests to determine the fatigue life. One was a reinforced bridge slab model (beam P3). P3 was pre-damaged and then reinforced with polyurethane concrete, and the fatigue performance of the reinforced bridge slab was studied by fatigue tests.

Based on the design drawings of the 16 m rib beam bridges in Standard Drawings for Highways and Bridges formulated by the Expert Committee of the Ministry of Transport of the People's Republic of China [9], the test model of the double T-beam bridge slabs was established in this paper. The model of bridge slabs and its transverse and longitudinal reinforcement are shown in Figures 1–3.

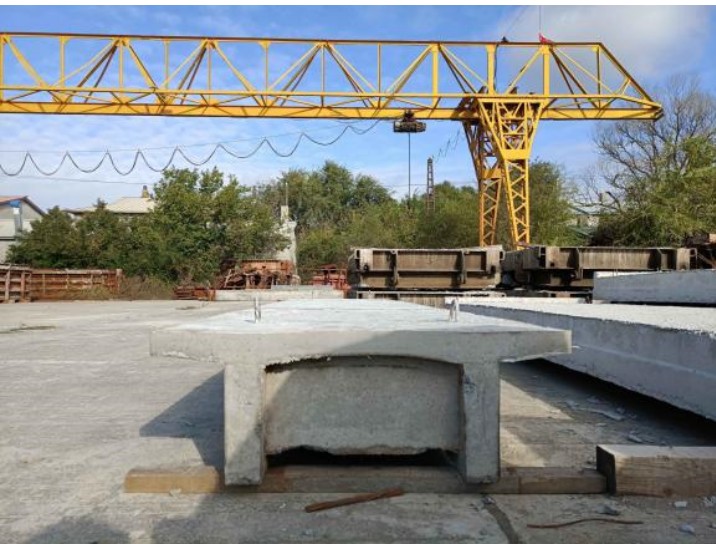

**Figure 1.** The model of the bridge slab.

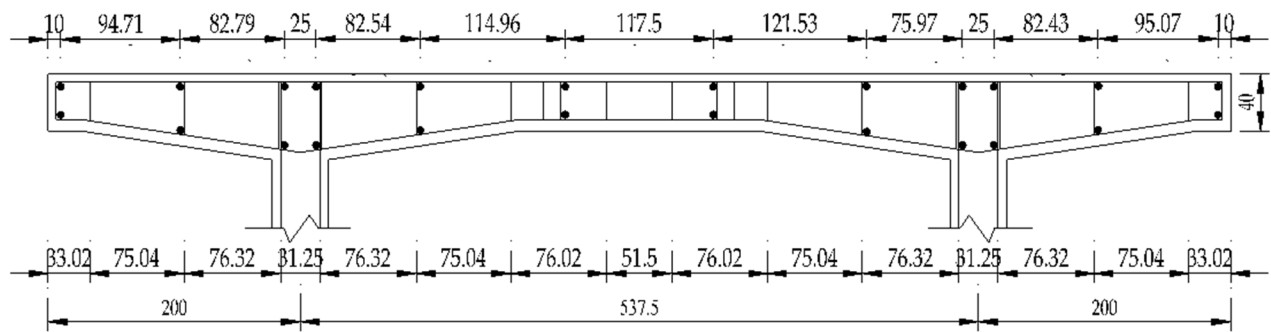

**Figure 2.** The transverse diagram of the bridge slab model (unit: mm).

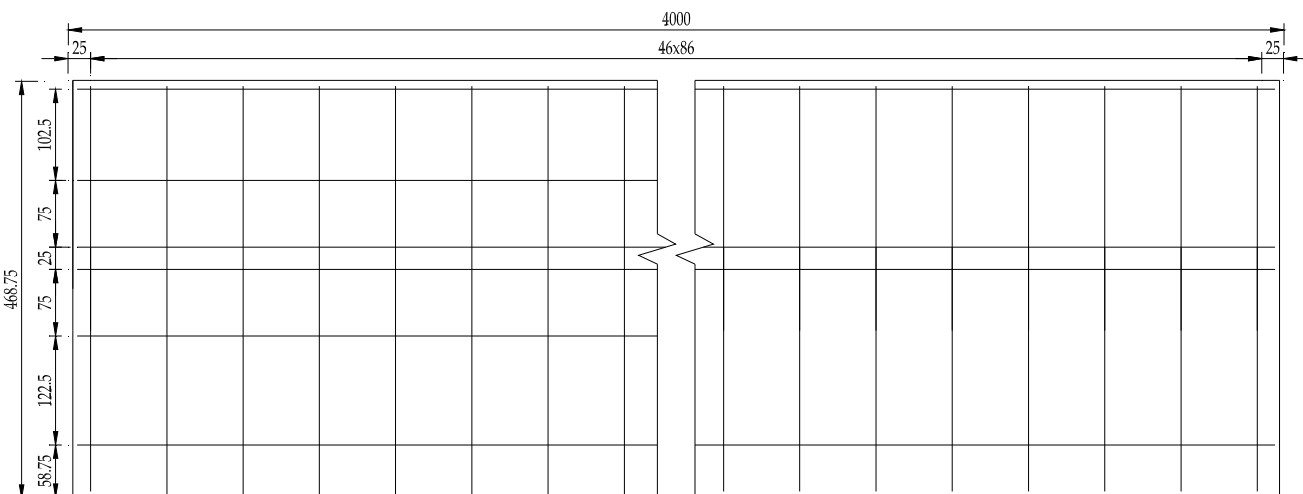

**Figure 3.** The longitudinal diagram of the bridge slab model (unit: mm).

Mechanical Indices of Materials

In the bridge slabs, the concrete strength grade was C40, and the steel bars were HRB400 steel bars, and the steel bar diameters were 6 mm and 10 mm. When pouring the test bridge slabs, the mechanical properties of the concrete and steel bars were tested according to the standard of the Ordinary Concrete Mechanical Properties Test Method

(GB/T 50081-2016) [10] and the Metal Tensile Test Method (GB 228.1-2010) [11]. The main mechanical properties of the concrete and steel bars on the test bridge slabs are shown in Table 1.

**Table 1.** The main mechanical properties of the concrete and steel bars on the test bridge slabs.

| Material varieties | Cubic Compressive Strength (MPa) | Prismatic Compressive Strength (MPa) | Yield Strength (MPa) |
|---|---|---|---|
| C40 concrete | 40.40 | 26.82 | - |
| 6 mm HRB400 | - | - | 420.46 |
| 10 mm HRB400 | - | - | 409.29 |

1. The dimensions of the cubic compressive strength specimens were 150 mm × 150 mm × 150 mm. 2. The dimensions of the prismatic compressive strength specimens were 150 mm × 150 mm × 300 mm.

### 2.1.2. Reinforcement of the Fatigue Damaged Test Slabs

The reinforcement material used in this study was polyurethane concrete, whose mixture ratio and mechanical properties are shown in Tables 2 and 3.

**Table 2.** The mixture ratio of polyurethane concrete.

| Cement | Fly Ash | Polyisocyanate | Modified Polyether Polyol | Water Reducer | Retarder |
|---|---|---|---|---|---|
| 1.20 | 0.40 | 1.00 | 1.00 | 0.15 | 0.05 |

**Table 3.** The mechanical properties of the polyurethane concrete.

| Density (g/cm$^3$) | Compressive Strength (MPa) | Tensile Strength (MPa) | Elastic Modulus (MPa) |
|---|---|---|---|
| 1.497 | 83 | 21.9 | 6721 |

Polyisocyanate and modified polyether polyols in raw materials can react with water to form lots of bubbles in the polyurethane concrete, which will reduce the mechanical properties and bonding properties of the polyurethane concrete. Therefore, the raw materials of the polyurethane concrete need to be strictly dewatered, and a water reducer was added in the mixing process. The final water content of the polyurethane concrete was less than 0.05%.

According to the standard of the Highway engineering cement and cement concrete test procedures (JTE E30-2005) [12], the workability of polyurethane concrete was tested. Under laboratory conditions of 20 °C, the initial setting time of the polyurethane concrete was 45 min, the final setting time was 135 min, and the maximum temperature of the polyurethane concrete in the reaction process was 30.1 °C. With the increase in the pouring molding time, the slump in the polyurethane concrete changed in four stages: From 0 to 45 min, the polyurethane concrete was liquid, and the slump decreased from 240 mm to 120 mm (the decreasing rate of 50.00%); from 45 to 90 min, the polyurethane concrete was in the plastic state, and the slump decreased from 120 mm to 70 mm (the decreasing rate of 41.76%); from 90 to 135 min, the polyurethane concrete was in the low plastic state, and the slump decreased from 70 mm to 50 mm (the decreasing rate of 28.75%); and when the time was greater than 135 min, the polyurethane concrete was in a dry dense state.

The reinforcement construction design was based on the principle that the ultimate load capacity of the damaged bridge slab after reinforcement should be restored to the original design ultimate load capacity. The longitudinal length of the reinforcement layer was 2775 mm, the maximum transverse width was 340 mm, and the maximum thickness was 25 mm. In addition, two longitudinal steel bars with a diameter of 3 mm and a spacing of 117.5 mm, and 33 transverse steel bars with a diameter of 3 mm and a spacing of 86 mm were arranged inside the reinforcement layer. During construction, planted bars 2 cm

higher than the bottom were arranged in the concrete at the bottom of the bridge slab and bound with steel mesh nodes in the reinforcement layer. The cross section and plan of the polyurethane concrete reinforcement layer are shown in Figures 4 and 5.

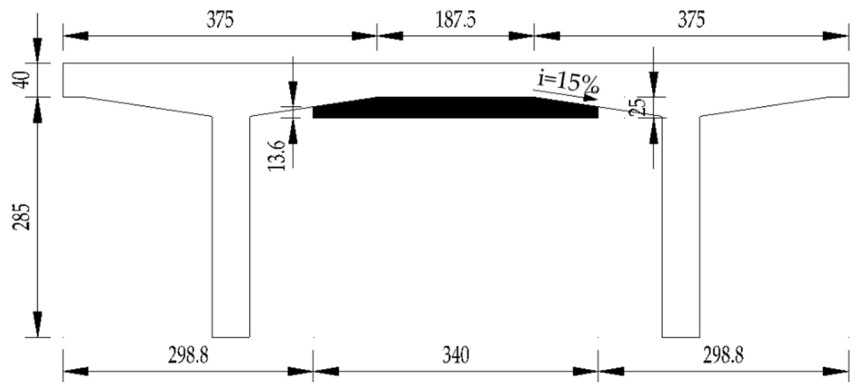

**Figure 4.** The cross-section of the polyurethane concrete reinforcement layer (unit: mm).

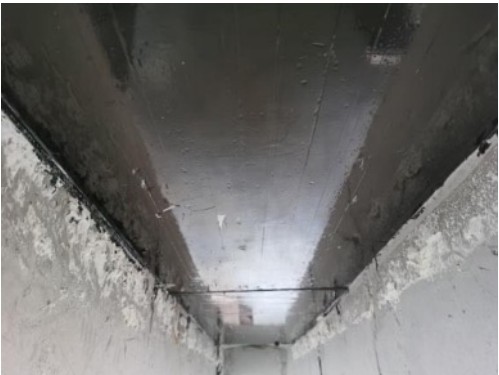

**Figure 5.** The plan of the polyurethane concrete reinforcement layer.

2.1.3. Reference Bridge Slab Test

Loading Scheme and Test Parameters Settings

First, static load tests were carried out on P1. A PLS-500 electro-hydraulic servo dynamic and static test machine (manufacturer DOCER, country China, city jinan in shandong province) was used as the loading device. Multi-stage loading and three-point bending were adopted in the test. The loading conditions at the testing site are shown in Figure 6.

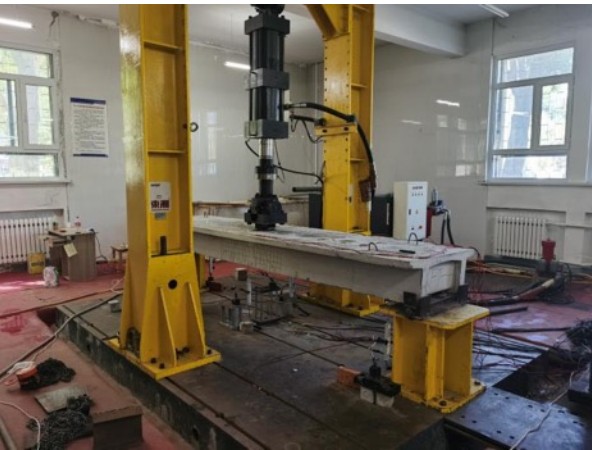

**Figure 6.** The loading conditions at the testing site.

Then, P2 was subjected to fatigue tests using the same loading device. According to the proportion of heavy vehicles announced in the Statistical Bulletin on the Development of Transportation Industry in 2019 [13] and the fatigue test results obtained by domestic and foreign scholars [5,14–16], the test fatigue load level was set too 0.65, which means that the ratio of the upper limit of the fatigue load to the ultimate load capacity of the bridge slab. Constant amplitude-controlled loading was adopted in the test, the waveform was a sine wave, and the fatigue loading frequency was 3.8 Hz. The lower and upper limits of the fatigue load were 15 kN and 75 kN, respectively. Deflection measuring points were arranged in the mid-span of the bridge slab, and reinforcement strain measuring points were arranged in the mid-span of the tensile reinforcement. When the fatigue cycles were 50,000, 100,000, 200,000, 500,000, 1 million, 1.5 million, and 2 million cycles, the static load tests were carried out on the bridge slab, where the load was the upper limit of the fatigue load. At this time, test data were collected, and then the loading was resumed.

Measuring Point Arrangement and Data Acquisition

In this paper, seven displacement measuring points were arranged at the middle span of the bridge slabs, the beam rib, and the middle line of the support. The displacement measurement range was 20 mm and the accuracy was 0.01 mm; the displacement meter at the beam rib and the middle line of the support was to eliminate the error caused by the vertical displacement of the bearing. Two strain measuring points were arranged on two 6 mm track slabs in the mid-span, and the steel strain gauge was 120-3AA. The arrangements of the deflection measuring points and reinforcement strain measuring points are shown in Figure 7.

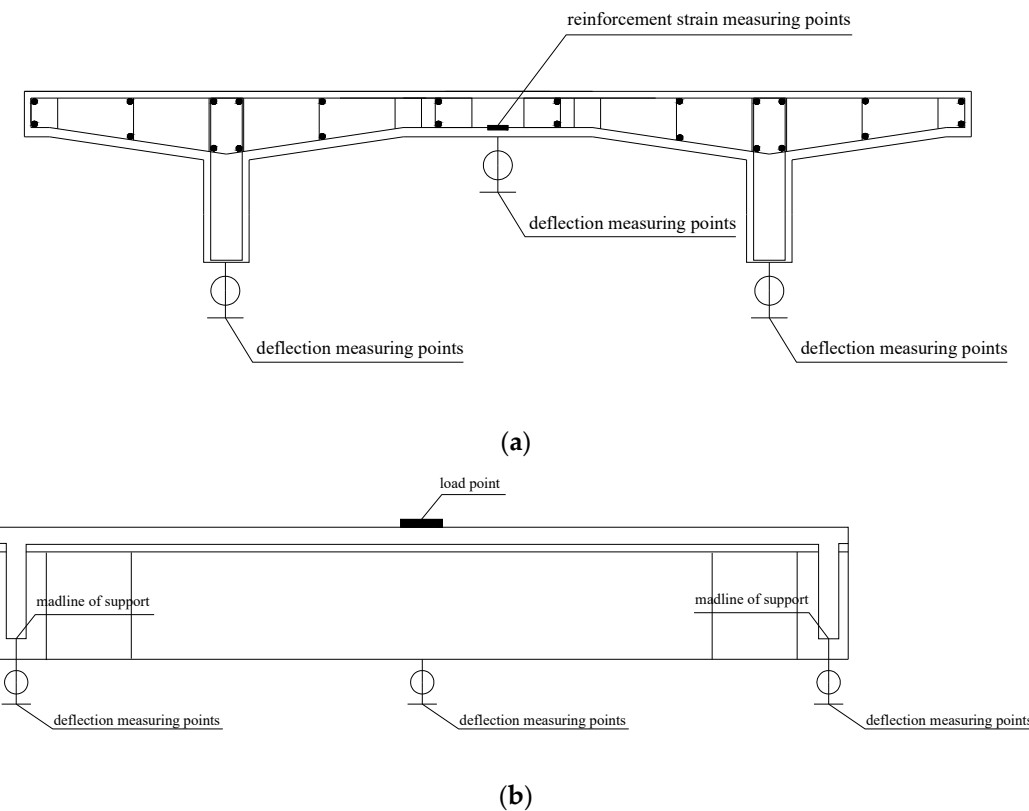

**Figure 7.** The arrangement of the deflection measuring and reinforcement strain measuring points. (**a**) Front view. (**b**) Side view.

In this experiment, a TDS-530 static data acquisition instrument was used for the data acquisition. When the fatigue number reached the specified value, the deflection of the

bridge slabs and the strain of steel bar were collected. The test data acquisition equipment are shown in Figure 8.

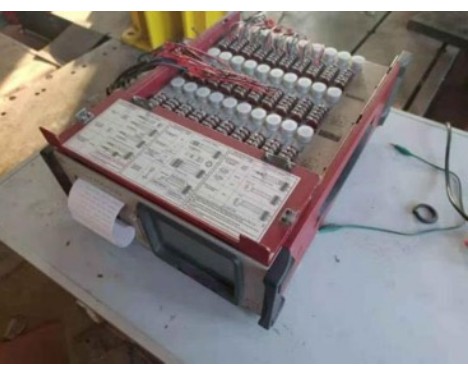

**Figure 8.** The test data acquisition equipment.

2.1.4. Fatigue Tests on the Reinforced Slab

Fatigue damage tests were carried out on P3. Constant amplitude-controlled loading was adopted in the test, the waveform was a sine wave, and the fatigue-loading frequency was 3.8 Hz. The lower and upper limits of the fatigue load were 15 kN and 75 kN, respectively. According to the P2 fatigue damage development, the fatigue damage of P3 was controlled in the steady development period of damage, and the fatigue cycles of P3 were finally determined to be 500,000 cycles. After damage, the maximum crack width of P3 at the slab bottom was 0.26 mm. According to the definition of damage degree (D) in the Palmgren–Miner linear damage degradation formula, $D = N_t/N_f$, where $N_t$ and $N_f$ are the cycles of the fatigue load and fatigue life of the bridge slab. Through the calculations, the damage degree of P3 was 0.3086, and its fatigue damage modes are shown in Figure 9.

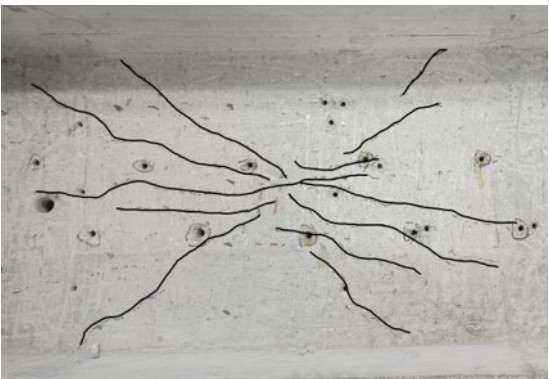

**Figure 9.** The fatigue damage mode of P3 after 500,000 fatigue cycles.

The damaged P3 beam was strengthened with polyurethane concrete as in Section 2.1.2. Under the fatigue cyclic load, the ultimate load capacity of the bridge slab will decrease due to fatigue damage. The purpose of reinforcing the bridge slab was to restore the ultimate load capacity of the damaged bridge slab to the original design value. Therefore, based on the design value of the ultimate load capacity of the bridge slab, the fatigue load parameters of the reinforced P3 were designed in this paper. The fatigue test parameters of the P2 bridge slab were formulated by the design value of the ultimate load capacity and the research results of previous researchers. In order to investigate the reinforcement effect of the polyurethane concrete and the fatigue performance of the reinforced bridge slab, the damaged P3 bridge slab after reinforcement used the same fatigue test parameters as P2. The lower and upper limits of the fatigue load were 15 kN and 75 kN, respectively, the waveform was a sine wave, and the fatigue loading frequency was 3.8 Hz

According to the Standard for the Test Methods of Long-term Performance and Durability of Ordinary Concrete [17], the test can be terminated if the specimen is not damaged after 2 million loads. To explore the fatigue performance of reinforced P3, the fatigue cycles were extended to 3 million cycles. The arrangement of measuring points and the fatigue cycles of collected data were the same as P2.

### 2.2. Fatigue Damage Finite Element Model

In repeated fatigue tests, the test results are often discrete. Therefore, based on fatigue experiments, this paper conducted a numerical analysis of the reinforced bridge slabs after damage based on the cumulative fatigue damage theory. Fatigue cumulative damage theory mainly studies the structural fatigue damage and its development law. Fatigue damage refers to the degree of continuous deterioration of materials caused by cyclic loading. When the material deteriorates to a critical value due to the cumulative fatigue damage, the material undergoes fatigue failure. Fatigue damage theory mainly solves three problems in fatigue numerical simulation:

1. Calculation of the material damage caused by the fatigue cyclic load;
2. Calculation of the material damage accumulation after the fatigue cycle load damage to the structure;
3. Calculation of the damage critical value of the material under fatigue failure.

In this paper, the finite element model of the reinforced bridge slabs after damage was established by the ABAQUS finite element analysis software. Based on the fatigue damage theory, the cumulative damage factor of the material constitutive relationship under fatigue cyclic loading was calculated. The fatigue damage of the structure was realized by reducing the strength and stiffness of the material, and the fatigue behavior of the reinforced bridge slabs after damage under different damage degrees was studied.

2.2.1. Establishment of the Constitutive Relationship

1. Constitutive relationship of concrete

According to Zhu [18], and the plastic damage model proposed by Lubliner et al. [19] and modified by Lee and Fenves [20], the fatigue damage constitutive relationship of concrete was determined based on the uniaxial tensile and compressive constitutive model of concrete in the Code for Design of Concrete Structures [21].

According to the results of the Special Group on Concrete Fatigue [22], the degraded elastic modulus of concrete can be calculated using Equation (1):

$$E_r = \left(1 - 0.33\frac{N_t}{N_f}\right)E_c \tag{1}$$

where $E_r$ represents the degraded elastic modulus of concrete; $E_c$ represents the initial elastic modulus of concrete; $N_t$ represents the cycles of fatigue load experienced by concrete; and $N_f$ represents the fatigue life of concrete.

According to the relevant data from the Special Group on Concrete Fatigue [22], the S–N relationship of the concrete fatigue life can be calculated using Equation (2):

$$\lg N_f = 14.7 - 13.5\left(\frac{\sigma_{max}}{f_{cu}}\right)^v \tag{2}$$

where $f_{cu}$ represents the standard value of concrete strength; $\sigma_{max}$ represents the peak strength of concrete; and $v$ represents the experimental coefficient, which is generally taken as 0.5.

According to Holmen [23], the residual fatigue strength of concrete can be calculated using Equation (3):

$$\sigma_r = f_{cu} - [f_{cu} - \sigma_{max}]\left(\frac{\lg(N_t)}{\lg(N_f)}\right)^v \tag{3}$$

where $\sigma_r$ represents the residual fatigue strength of the concrete after fatigue cycles of $N_t$; $f_{cu}$ represents the standard value of the concrete strength; and $v$ represents the experimental coefficient. According to Meng's test [24], 2.055 was taken for the compressive tests, and 2.514 for the tensile tests.

The constitutive model of concrete can be calculated according to Equations (4)–(12), and the stress-strain relationship of concrete is shown in Figure 10.

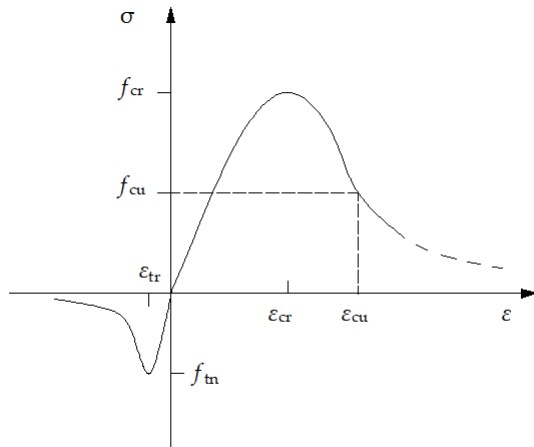

**Figure 10.** The stress–strain relationship of concrete.

The compressive constitutive model:

$$\sigma = k_c E_r \varepsilon \tag{4}$$

$$k_c = \begin{cases} \dfrac{n\rho_c}{n-1+x_c^n}, & x_c \leq 1 \\ \dfrac{\rho_c}{\alpha_c(x-1)^2 + x_c}, & x_c > 1 \end{cases} \tag{5}$$

$$\rho_c = \frac{f_{cu}}{E_N \varepsilon_{cr}} \tag{6}$$

$$x_c = \frac{\varepsilon - \varepsilon_{fr}(N_t)}{\varepsilon_{cr}} \tag{7}$$

$$n = \frac{E_N \varepsilon_{cr}}{f_{cu}} \tag{8}$$

The tensile constitutive model:

$$\sigma = k_t E_r \varepsilon \tag{9}$$

$$k_t = \begin{cases} \rho_t \left(1.2 - 0.2x_t^5\right), & x_t \leq 1 \\ \dfrac{\rho_t}{\alpha_t(x-1)^{1.7} + x_t}, & x_t > 1 \end{cases} \tag{10}$$

$$\rho_t = \frac{f_{tn}}{E_N \varepsilon_{tr}} \tag{11}$$

$$x_t = \frac{\varepsilon - \varepsilon_{tr}(N_t)}{\varepsilon_{tr}} \tag{12}$$

where $\sigma$ and $\varepsilon$ represent the tensile (compressive) stress and strain of concrete, respectively; $\alpha_c$ and $\alpha_t$ represent the shape parameters of the descending section of the concrete curve determined according to the Code for Design of Concrete Structures [21]; $\varepsilon_{fr}(N)$ represents the residual strain of concrete after fatigue cycles of $N_t$, and $f_{cu}, f_{tn}, \varepsilon_{cr}$, and $\varepsilon_{tr}$ represent the corresponding residual compressive strength, tensile strength, peak compressive, and tensile strains of concrete after fatigue cycles of $N_t$.

2.    Constitutive relationship of steel bars

According to Meng [25] and relevant data from Zeng and Li [26], the S–N relationship of the fatigue life of steel bars is calculated using Equation (13):

$$\lg N_f = 44.4580 - 18.3614 \lg(\Delta\sigma) \tag{13}$$

where $N_f$ represents the fatigue life, and $\Delta\sigma$ represents the stress amplitude.

According to Zhu [26], the fatigue damage of the elastic modulus of steel bars was not considered in this paper. The residual fatigue strength of the steel bars can be calculated using Equation (14):

$$\sigma_r = f_y \left[ 1 - \frac{\lg(N_t)}{\lg(N_f)} \left( 1 - \frac{\sigma_{max}}{\sigma_\gamma} \right) \right] \tag{14}$$

where $\sigma_\gamma$ represents the yield strength of the steel bars.

According to Li [27], under the action of high-cycle fatigue, the stress level of the steel bar is usually low, its stress and deformation are basically in the elastic stage, and the steel bar will not reach the yield strength. The fatigue failure of steel bars is a transient fracture due to the failure of the residual effective bearing area under tensile load. In addition, Zhu [26] and Zhu and Yan [28] used the bilinear elastic–plastic model of reinforcement to simulate the fatigue damage of the reinforced concrete rectangular beam and bridge deck, respectively. The results showed that the simulated value was in good agreement with the test value curve, and the maximum error was less than 10%.

Therefore, in this paper, according to the above parameters proposed by the Special Group on Concrete Fatigue [22] and the research results in the literature, the bilinear elastic-plastic model was used to calculate the fatigue damage stress-strain relationship of the reinforcement under different damage states. The constitutive model of the steel bar under fatigue damage is shown in Figure 11.

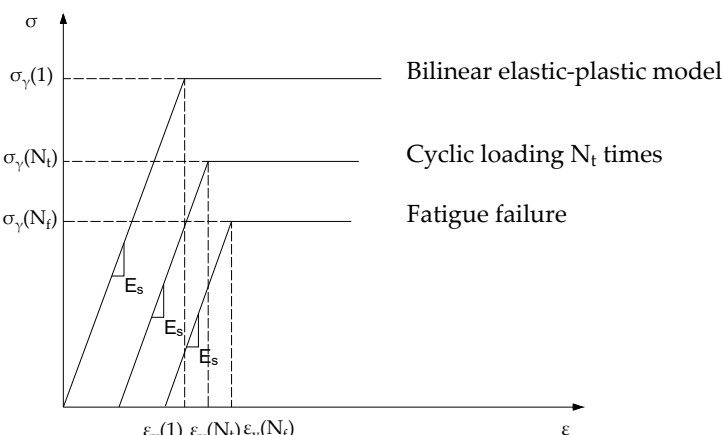

**Figure 11.** The constitutive model of the steel bar under fatigue damage.

The stress–strain relationship of the steel bars under different damage states can be calculated using Equations (15)–(17):

The residue strain:

$$\Delta\varepsilon_r(N_t - 1) = \frac{[\sigma_r(N_t) - \sigma_r(N_{t-1})]}{E_s} \tag{15}$$

The yield strain:

$$\varepsilon_y(N_t) = \Delta\varepsilon_r(N_t - 1) + \frac{\sigma_r(N_t)}{E_s} \tag{16}$$

The stress–strain relationship of the steel bars:

$$\sigma(N_t) \begin{cases} E_s\varepsilon(N_t) & (\Delta\varepsilon_r(N_t - 1) < \varepsilon(N_t) \leq \varepsilon_y(N_t)) \\ \sigma_{max}[1 - \frac{\lg(N_t)}{\lg(N_f)}\left(1 - \frac{\sigma_{max}}{\sigma_\gamma}\right) & (\varepsilon(N_t) > \varepsilon_y(N_t)) \end{cases} \tag{17}$$

3. Constitutive relationship of polyurethane concrete

According to Zhang [29], polyurethane concrete has a good anti-fatigue performance, and no obvious fatigue damage was found in the process of the fatigue loading of polyurethane concrete. Therefore, the constitutive relationship of the polyurethane concrete in this paper did not consider fatigue damage. The stress–strain relationship of the polyurethane concrete adopted the measured curve, and the uniaxial stress–strain curve of the polyurethane concrete is shown in Figure 12.

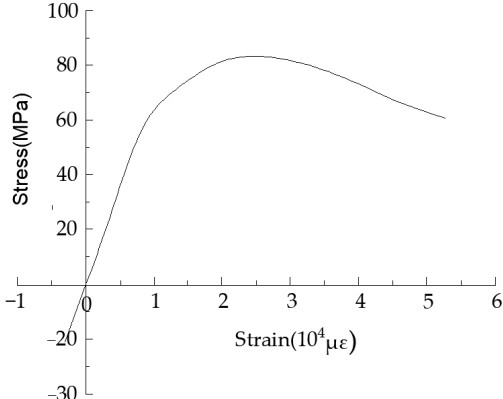

**Figure 12.** The uniaxial stress–strain curve of the polyurethane concrete.

In the initial loading stage, the compressive stress–strain curve developed from linear to nonlinear, and the peak compressive strength and strain were 83 MPa and 25,406 $\mu\varepsilon$, respectively. In the later loading stage, the compressive stress–strain curve was nonlinear, and the strain at failure was 52,776 $\mu\varepsilon$. The tensile stress–strain curve was almost linear, and the peak tensile strength and strain were 21.9 MPa and 3258 $\mu\varepsilon$, respectively. The relationship curve was fitted, and the constitutive relationship of the polyurethane concrete can be calculated using Equations (18)–(21):

The compressive elastic section ($0\ \mu\varepsilon < \varepsilon \leqq 8644\ \mu\varepsilon$):

$$\begin{aligned} \sigma &= 0.0067\varepsilon + 0.2399 \\ R^2 &= 0.9998 \end{aligned} \tag{18}$$

The compressive plastic section ($8644\ \mu\varepsilon < \varepsilon \leqq 25{,}406\ \mu\varepsilon$):

$$\begin{aligned} \sigma &= -1 \times 10^{-7}\varepsilon^2 + 0.0051\varepsilon + 22.255 \\ R^2 &= 0.9996 \end{aligned} \tag{19}$$

The compressive descending section ($25{,}406\ \mu\varepsilon < \varepsilon \leqq 52{,}776\ \mu\varepsilon$):

$$\begin{aligned} \sigma &= -1 \times 10^{-8}\varepsilon^2 - 7 \times 10^{-5}\varepsilon + 91.184 \\ R^2 &= 0.9995 \end{aligned} \tag{20}$$

The tensile linear section ($-3258\ \mu\varepsilon < \varepsilon \leqq 0$):

$$\begin{aligned} \sigma &= 0.0067\varepsilon - 0.2708 \\ R^2 &= 0.9987 \end{aligned} \tag{21}$$

2.2.2. Establishment of the Model

The finite element software ABAQUS was used to establish the model. The T3D2 truss element was used to simulate the steel bars and the C3D8R solid element was used to simulate the concrete and polyurethane concrete. Embedded regions were used between the bridge slab, polyurethane concrete reinforcement layer, and their internal steel meshes,

and the polyurethane concrete reinforcement layer was tied with the bridge slab. The experimental boundary conditions of the bridge slab in the actual tests were numerically simulated: One end restrained the longitudinal displacement, and the other end restrained the longitudinal and vertical displacement. Based on the damage theory, the fatigue damage was regarded as the cumulative factor of the constitutive damage, and the fatigue damage was simulated by reducing the strength and stiffness of the material after a certain number of fatigue cycles. The mesh element in the geometric model of the bridge slab and the model of the polyurethane concrete reinforcement layer are shown in Figures 13 and 14.

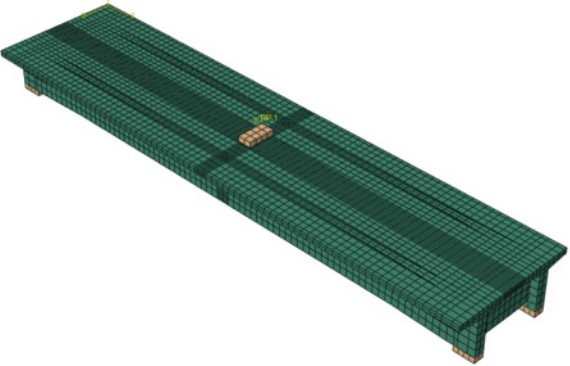

**Figure 13.** The mesh element in the geometric model of the bridge slab.

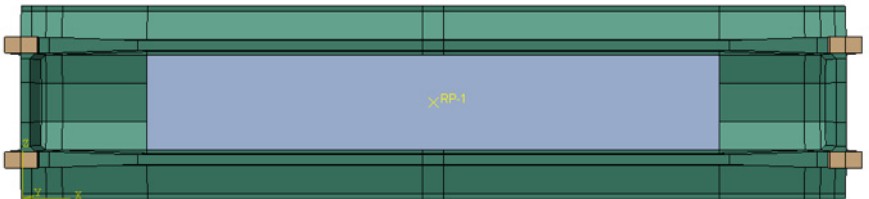

**Figure 14.** The model of the polyurethane concrete reinforcement layer.

## 3. Results

### 3.1. Tests on the Reference Slabs

The test results of the reference bridge slab P1 were as follows. The damage mode at the slab bottom of the P1 bridge slab under static loads is shown in Figure 15.

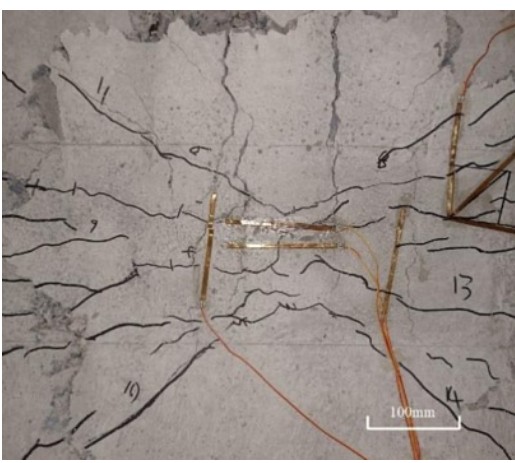

**Figure 15.** The damage mode at the slab bottom of P1 under static loads.

When the load reached 35 kN, longitudinal cracks appeared at the slab bottom. When it reached 80 kN, scattered oblique cracks appeared at the slab bottom, and the maximum

crack width reached 0.21 mm. When the load further increased to 115 kN, the cracks at the slab bottom developed rapidly, and the maximum crack width reached 1.4 mm. Ultimately, the punch failure occurred on the bridge slab.

The test result of reference bridge slab P2 were as follows. The damage mode at the slab bottom of P2, the relationship between the mid-span deflection and fatigue cycles, and the relationship between the reinforcement strain and fatigue cycles of P2 are shown in Figures 16–18.

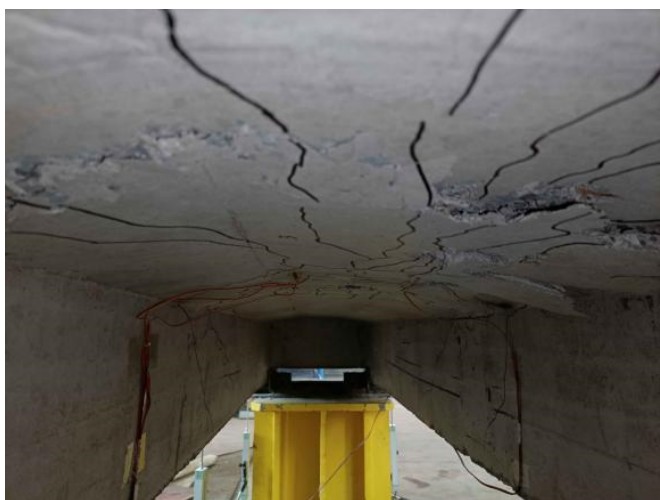

**Figure 16.** The fatigue damage mode at the slab bottom of P2.

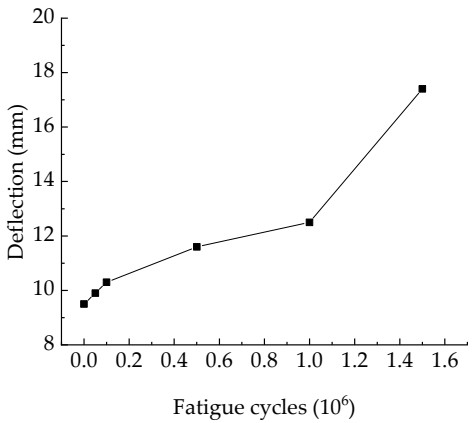

**Figure 17.** The relationship between the mid-span deflection and fatigue cycles of P2.

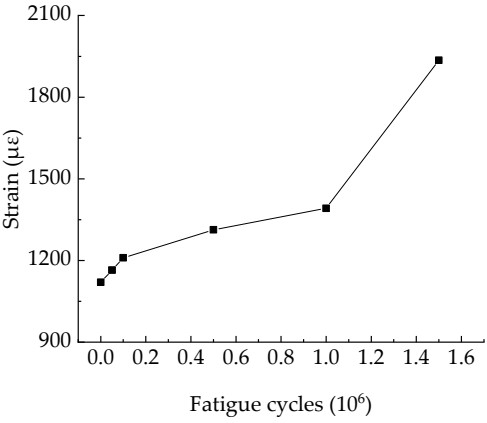

**Figure 18.** The relationship between the reinforcement strain and fatigue cycles of P2.

The results showed that the development of the P2 fatigue damage can be divided into three stages. In the first 100,000 cycles, the bridge slab was in the initial stage of fatigue damage, and no macro fatigue cracks were found at the slab bottom. With the increase in fatigue cycles, the mid-span deflection and reinforcement strain gradually increased, and the mid-span deflection reached 10.30 mm. From 100,000 to 1 million cycles, the bridge slab was in the period of the steady development of fatigue damage. At 100,000 cycles, longitudinal cracks began to appear at the bottom of the slab, and the maximum crack width was 0.18 mm. With the increase in fatigue cycles, the number, length, and width of the fatigue cracks continued to develop. At this time, there was no significant change in the mid-span deflection and the slope of the reinforcement strain curve. At 1 million cycles, the mid-span deflection increased to 12.52 mm, and the maximum crack width was 0.36 mm. From 1 million to 1.5 million cycles, the bridge slab was in a period of accelerated development of fatigue damage. Scattering cracks began to appear at the bottom of the plate. With the increase in the fatigue cycles, the cracks developed rapidly. At this time, the slope of the mid-span deflection and reinforcement strain curve suddenly increased. At 1.5 million cycles, the mid-span deflection increased to 17.40 mm, and the maximum crack width reached 1.2 mm. When the bridge slab reached the fatigue life (1.62 million cycles), the fatigue punch failure occurred on the bridge slab.

### 3.2. Fatigue Tests on the Reinforced Slab

The test result of the reference bridge slab reinforced P3 were as follows. The reinforcement layer at the slab bottom of the reinforced P3 after 3 million fatigue cycles, the relationship between the mid-span deflection and fatigue cycles, and the relationship between the reinforcement strain and the fatigue cycles of P3 are shown in Figures 19–21.

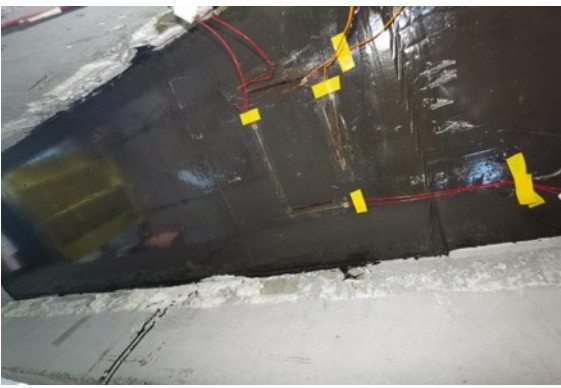

**Figure 19.** The reinforcement layer at the slab bottom of the reinforced P3 after 3 million fatigue cycles.

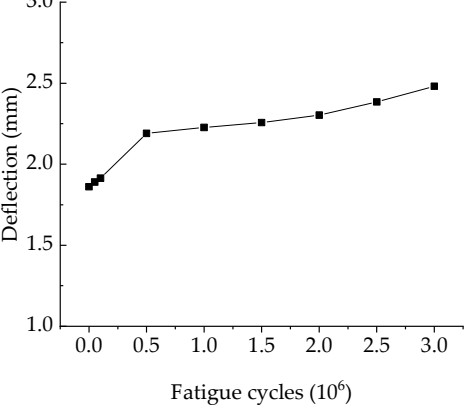

**Figure 20.** The relationship between the mid-span deflection and fatigue cycles of the reinforced P3.

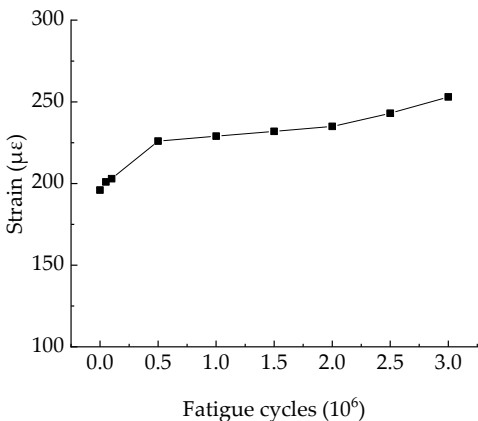

**Figure 21.** The relationship between the reinforcement strain and fatigue cycles of the reinforced P3.

The test phenomena results showed that the higher stiffness and lower elastic modulus of the polyurethane concrete could effectively improve the energy absorption capacity of the structure, which significantly reduced the fatigue damage of the structure under fatigue cyclic loading. Therefore, after 3 million cycles of the fatigue load, the reinforced P3 did not show the occurrence of fatigue failure. Moreover, during the test, there was no interface separation between the polyurethane concrete reinforcement layer and the bottom of the bridge slab. This shows that the polyurethane concrete had good mechanical properties, and its bonding with concrete was also very excellent, which is a good reinforcement material.

Figures 20 and 21 show that the development of the reinforced P3 fatigue damage can be divided into two stages. In the first 500,000 cycles, the bridge slab was in the initial stage of fatigue. With the increase in fatigue cycles, the mid-span deflection and reinforcement strain gradually increased. The end of this period had been delayed by 400,000 cycles compared to P2. At 500,000 cycles, the mid-span deflection of the reinforced P3 was 2.19 mm, which was 78.73% lower than that of P2 in the same period, and the mid-span deflection of the reinforced P3 was 0.33 mm compared to the initial value (a growing rate of 17.74%) and decreases by 13.83% compare with P2 in this period. From 500,000 to 3 million cycles, the bridge slab was in the period of a steady stage of fatigue damage. With the increase in fatigue cycles, the increase in the mid-span deflection and reinforcement strain was not obvious, and the duration of this stage increased significantly compared with P2. At 3 million cycles, the mid-span deflection of the reinforced P3 was 2.48 mm, which was 81.6% lower than that of P2 in the same period, and the mid-span deflection of the reinforced P3 was 0.62 mm compared to the initial value (a growing rate of 33.31%) and decreased by 49.83% compared with P2 in this period.

To sum up, polyurethane concrete plays a positive role in improving the anti-fatigue performance of a bridge slab. During the whole process of the fatigue tests, no fatigue damage occurred on the bridge slab, and no fatigue cracks occurred on the reinforced polyurethane layer. The fatigue life of the reinforced bridge was also greatly increased. Moreover, similar to P2, the fatigue damage of the reinforced P3 also developed in stages. However, under the same fatigue load, compared with P2, the boundary point of each damage period of the reinforced P3 lagged, and the mid-span deflection and fatigue damage decreased significantly in each damage period.

The relationship between the increase in the mid-span deflection and fatigue cycles of P2 and the reinforced P3 is shown in Figure 22.

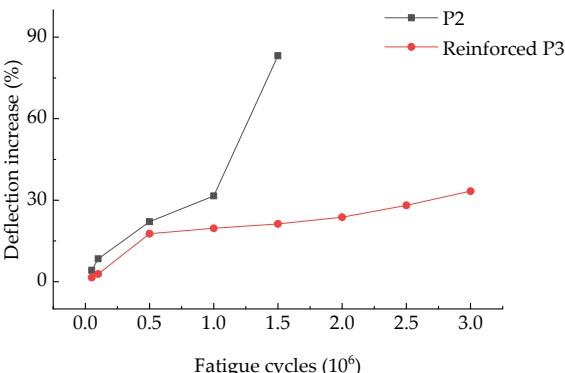

**Figure 22.** The relationship between the increase in the mid-span deflection and fatigue cycles of the P2 and reinforced P3.

*3.3. Finite Element Simulation Results*

3.3.1. Verification of the Finite Element Model

Under the test condition of the reinforced P3, the fatigue damage of the reinforced bridge slab was simulated. At the initial stage of fatigue, the fatigue damage of the bridge slab was small, and there was no obvious difference when compared with the undamaged bridge slab. Therefore, the simulation in this paper started from 500,000 cycles. The simulated mid-span deflection obtained from the finite element model was compared with the measured deflection obtained from the actual tests to verify the validity of the finite element analysis. The comparison of the deflection between the measured and simulated values is shown in Figure 23.

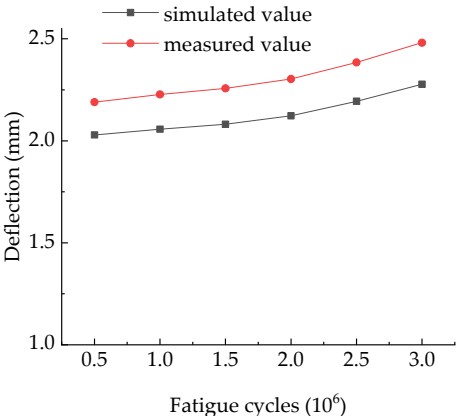

**Figure 23.** The comparison of the deflection between the measured and simulated values.

The results showed that the development trend of the fatigue damage in the numerical simulation was consistent with that in the actual tests, which is in line with the two-stage development law of the fatigue damage in the actual tests when the fatigue cycles ranged from 500,000 to 3 million cycles. When the fatigue cycles reached 3 million, the difference between the simulated deflection (2.277 mm) and the experimental deflection (2.48 mm) reached the maximum, and the relative error was 8.95%, which was within the acceptable range. Thus, the validity of the finite element model proposed in this paper was verified.

3.3.2. Fatigue Performance Analysis of Bridge Slabs Reinforced with Polyurethane Concrete under Different Damage Degrees

In this simulation, the fatigue load level was 0.65. According to the three stages of the P2 fatigue damage development, the characteristic points corresponding to each period were selected and calculated according to the calculation method of damage degree in Section 3.2. Ultimately, three damage degrees were selected, namely 0.0309(S1), 0.3086(S2),

and 0.9259(S3), and six fatigue cycles were considered under each damage degree, which were 500,000 cycles, 1 million cycles, 1.5 million cycles, 2 million cycles, 2.5 million cycles, and 3 million cycles.

The damage nephogram of the S1 model after 2 million fatigue cycles, the relationship between the residual ultimate load capacity and damage degree, and the relationship between the mid-span deflection and damage degree are shown in Figures 24–26.

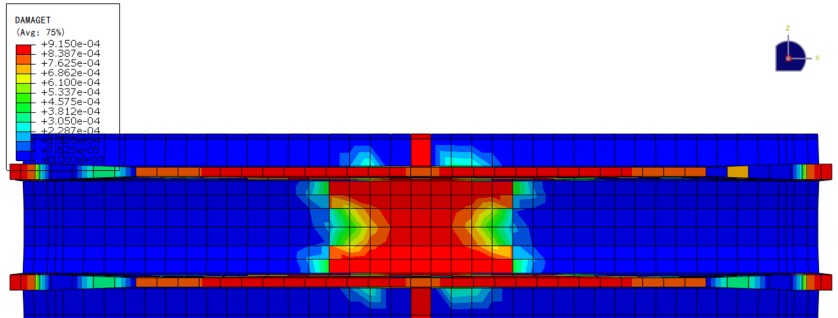

**Figure 24.** The damage nephogram of the S1 model after 2 million fatigue cycles.

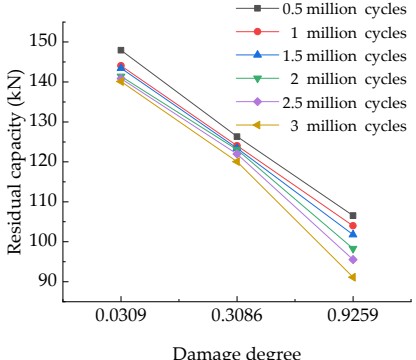

**Figure 25.** The relationship between the residual ultimate load capacity and damage degree.

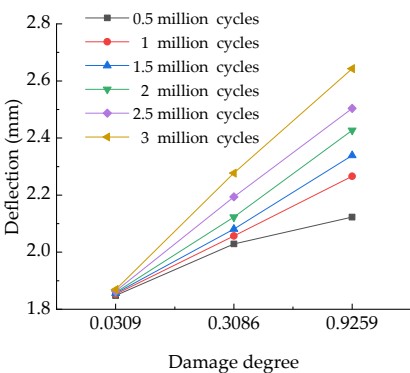

**Figure 26.** The relationship between the mid-span deflection and damage degree.

The results showed that the anti-fatigue performance of the reinforced bridge slab decreased with the increasing damage degree. From 500,000 to 3 million cycles, the fatigue damage of the bridge slab under the S1 damage degree was relatively slight, the residual ultimate load capacity decreased from 147.952 kN to 140.133 kN (a decreasing rate of 5.28%), and the deflection increased from 1.848 mm to 1.869 mm. The fatigue damage of the bridge slab under the S2 damage degree was roughly the same as that of S1, the residual ultimate load capacity decreased from 126.313 kN to 120.054 kN (a decreasing rate of 4.96%), and the deflection increased from 2.029 mm to 2.277 mm. In addition, compared with S1, the

residual ultimate load capacity decreased by 14.33%, and the mid-span deflection increased by 21.83%. The fatigue damage of the bridge slab under the S2 damage degree was the most obvious, the residual ultimate load capacity decreased from 106.534 kN to 91.125 kN (a decreasing rate of 14.46%), and the deflection increased from 2.123 mm to 2.643 mm. Moreover, compared with S1, the residual ultimate load capacity decreased by 34.97%, and the mid-span deflection increased by 41.41%; compared with S2, the residual ultimate load capacity decreased by 24.10%, and the mid-span deflection increased by 16.07%.

When reaching the normal service life of the bridge slab (2 million cycles), the residual ultimate load capacity of the reinforced bridge slabs under the S1, S2, and S3 damage degrees were 141.3914 kN, 123.0626 kN, and 98.2851 kN. Although the residual ultimate load capacity of the bridge slab under the S3 damage degree was lower than the design ultimate load capacity of the bridge slab (115 kN), it was still higher than the upper limit of the fatigue load (75 kN) under the fatigue level of 0.65, proving that the reinforced bridge slabs meet the requirements of daily use.

### 3.3.3. Prediction Model of the Residual Ultimate Load Capacity of Damaged Bridge Slabs Reinforced with Polyurethane Concrete

The rib beam bridge occupies a large proportion of the completed small- and medium-span bridges. Under the long-term heavy load and overload of vehicles, the ultimate load capacity of many rib beam bridge slabs was reduced due to the fatigue damage and appeared fatigue damage. In order to quantify the development law of fatigue damage of the damaged bridge slabs reinforced with polyurethane concrete and provide theoretical support for the reinforcement and repair of a large number of fatigue damaged bridge slabs, the residual ultimate load capacity of the reinforced bridge slabs under three different damage degrees in Section 3.3.2 was analyzed by the least square method. The prediction model of the residual ultimate load capacity of the reinforced bridge slabs was established. The relationship between the damage degree and the residual ultimate load capacity under different fatigue cycles is shown in Figure 27.

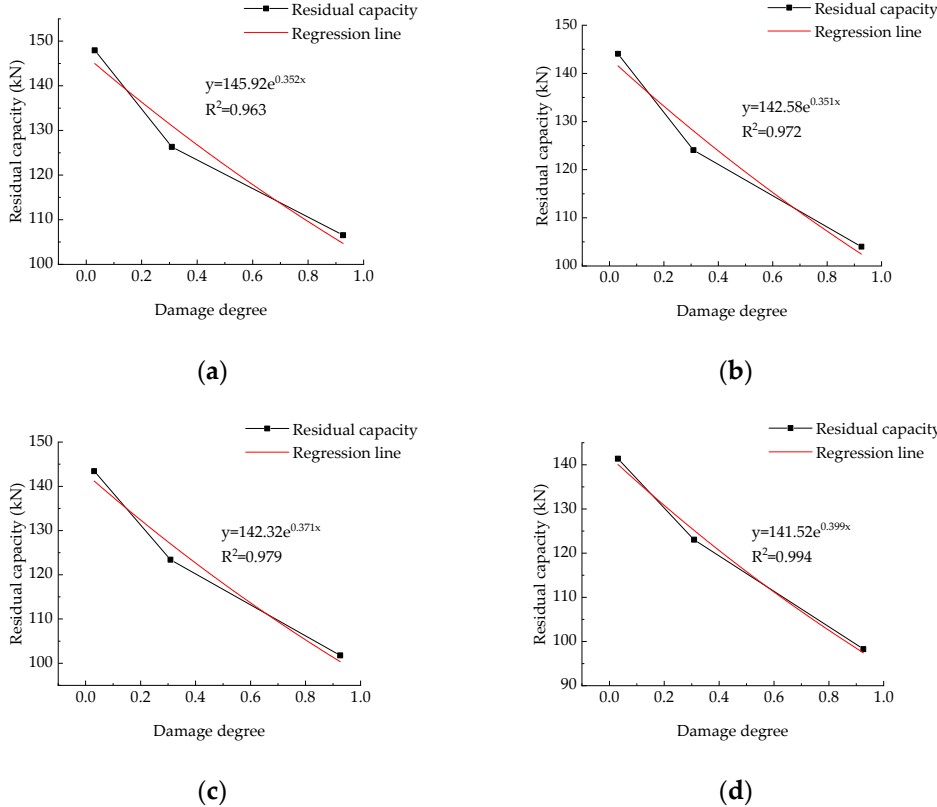

**Figure 27.** *Cont.*

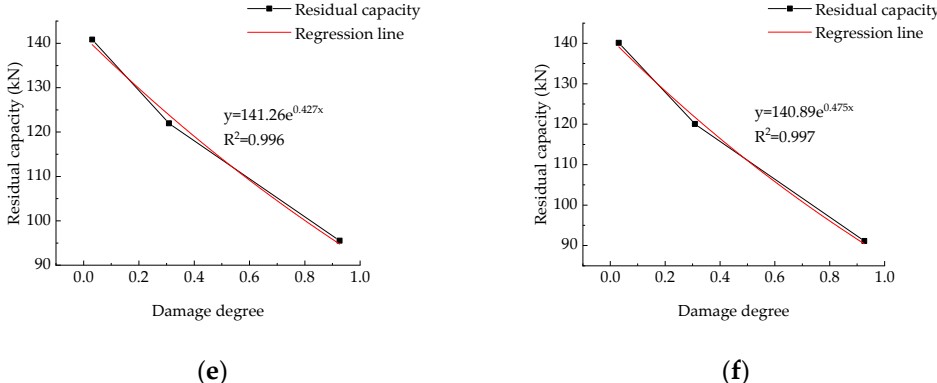

**Figure 27.** The relationship between the damage degree and the residual ultimate load capacity under different fatigue cycles. (**a**) 0.5 million cycles; (**b**) 1 million cycles; (**c**) 1.5 million cycles; (**d**) 2 million cycles; (**e**) 2.5 million cycles; (**f**) 3 million cycles.

The results showed that there is an exponential relationship between the damage degree and residual ultimate load capacity under different fatigue cycles. The regression equation between the residual ultimate load capacity and damage degree was established

$$F = ae^{bD} \tag{22}$$

where coefficients a and b are related to the fatigue cycles; F represents the residual ultimate load capacity of the damaged bridge slab after reinforcement; and D represents the damage degree of the bridge slab before reinforcement.

Regression analysis was performed on coefficients a and b under different fatigue cycles, and Equations (23) and (24) were obtained. The relationship between the different coefficients (a and b) and fatigue cycles is shown in Figure 28.

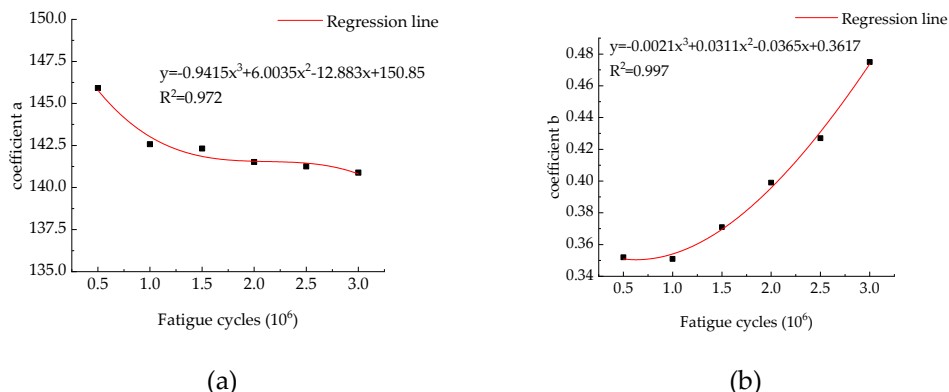

**Figure 28.** The relationship between the different coefficients (a and b) and fatigue cycles. (**a**) Coefficient a. (**b**) Coefficient b.

Intervention studies involving animals or humans and other studies that require ethical approval must list the authority that provided the approval and the corresponding ethical approval code.

$$a = -0.9415S^3 + 6.0035S^2 - 12.883S + 150.85$$
$$R^2 = 0.972 \tag{23}$$

$$b = -0.0021S^3 + 0.0311S^2 - 0.0365S + 0.3617$$
$$R^2 = 0.997 \tag{24}$$

where S represents the fatigue load cycles.

By substituting Equations (23) and (24) into Equation (22), it can be obtained that the calculation formula (25) of the residual ultimate load capacity of the reinforced bridge slab corresponding to any fatigue cycles and damage degree can be obtained.

$$F = \left(-0.9415S^3 + 6.0035S^2 - 12.883S + 150.85\right)e^{(-0.0021S^3 + 0.0311S^2 - 0.0365S + 0.3617)D} \tag{25}$$

**4. Conclusions**

1.  The fatigue damage development process of the reference bridge slab can be divided into three stages: The initial development of fatigue damage, the stable development period of fatigue damage, and the accelerated development period of fatigue damage. When the bridge deck reached the fatigue life (1.62 million cycles), the punch failure occurred on the bridge slab, and the structure lost its ultimate load capacity.

2.  Polyurethane concrete is beneficial in improving the anti-fatigue performance of bridge slabs. After experiencing the same fatigue load of 3 million cycles, fatigue damage of the damaged reinforced bridge slabs did not occur. The fatigue damage of the reinforced bridge slabs also developed in stages. Compared to the unreinforced bridge slab, each stage of fatigue damage lagged, and the fatigue damage in each stage was significantly reduced.

3.  Based on the simulation results of ABAQUS, the anti-fatigue performance of the reinforced bridge slab decreased with the increasing damage degree. Furthermore, a residual-bearing-capacity model including the damage degree and fatigue cycles of the reinforced bridge slabs was proposed.

**Author Contributions:** Conceptualization, T.Y. and Y.W. (Yifan Wang); Methodology, T.Y. and L.Y.; Software, L.Z. and Y.W. (Yifan Wang).; Validation, B.C.; Formal analysis, Y.W. (Yifan Wang); Investigation, Y.W. (Yifan Wang) and L.Z.; Data curation, Y.W.(Yifan Wang); Writing—review and editing, Y.W. (Yifan Wang), T.Y., and L.Y.; Visualization, Y.W. (Yuxuan Wu) and B.C.; Supervision, T.Y.; Project administration, Y.W. (Yuxuan Wu). All authors have read and agreed to the published version of the manuscript.

**Funding:** The authors gratefully appreciate the support from our various partners. This study was supported by the Department of Communications of Heilongjiang Province (grant no. 2019-034); the 2nd Batch of Industry–University Cooperation Collaborative Education Project in 2021 by the Department of Higher Education of Ministry of Education (grant no. 202102478002, Lihui Yin, School of Civil Engineering, Heilongjiang University); and the Natural Disaster Survey and Scientific Research Project of Heilongjiang Province (Lihui Yin, School of Civil Engineering, Heilongjiang University).

**Institutional Review Board Statement:** Not applicable.

**Informed Consent Statement:** Not applicable.

**Data Availability Statement:** The data used to support the findings of this study are available from the corresponding author upon request.

**Conflicts of Interest:** The authors declare no conflict of interest.

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
