# Peer review of "Fatigue Performance of Rib Beam Bridge Slabs Reinforced with Polyurethane Concrete Based on the Damage Theory"

_buildings, doi:10.3390/buildings12060704_

Round 1

Reviewer 1 Report

The article is devoted to the study of a cyclic fracture of concrete slabs and the influence on this process of reinforcement with a layer of polyurethane concrete. The authors investigated rib beam bridge slabs. Fatigue tests in the initial state, in the damaged state after preloading and after repair were carried out. Numerical modeling of the slabs has been done and a model of residual bearing capacity has been proposed, including the damage degree and the number of fatigue cycles of reinforced bridge slabs.

The relevance of this work is related to the requirement to ensure safe operation of bridges after their prolonged operation, which causes the damage development and reduction of bearing capacity.

Broad comments

Adding a reinforcing layer of polyurethane concrete increased the thickness of the slab by 62%, while the cyclic loads remained the same as for the slab with the original cross-section. Therefore, I think it is inaccurate to directly compare the fatigue test results of slabs P2 and P3. In my opinion, the authors should have increased the cyclic load on the reinforced slab in proportion to the increased slab cross-section.

The relationship between the load and damage obtained by the authors in this form makes it difficult to use in practice because it is suitable for describing the results of only one type of slab studied by the authors. If other slab is taken, the loads will be different, and this model will not work. Perhaps the stress or local deformations should have been correlated with the damage parameter.

As a wish, the authors could have supplemented these experiments with acoustic emission registration and evaluation of the deformation fields by digital image correlation. This would provide more information on the fracture stages of concrete slabs.

Specific comments on the article

  • Line 125: replace "three-point loading" with "three-point bending".
  • Line 216: Replace "Figs. 6" to "Fig. 6".
  • Line 132 In the phrase "the test fatigue load level was set as 0.65" it is not clear from which load 0.65 is taken?
  • Comments on the whole article: it is more appropriate to write common notation "cycles" instead of "times" for description of fatigue experiments.
  • 11. The yield strength is usually denoted as σY. In addition, the model of deformation of steel rods used is too simplified, which calls into question the adequacy of the simulation. In my opinion, a deformation curve similar to the one in Fig. 12 should have been used.
  • Mistake in numeration of formulas. It is necessary to check up by the paper text.
  • In my opinion, the scale in Fig. 15 should have been put for clarity. In the presented form, it is not clear what is the size of the cracks in the figure.

Reviewer 2 Report

Dear Authors,

You can find my comments below:

  1. The experimental study should be explained in the abstract.
  2. In Table 1, the dimensions of the cube and prism specimens should be given. Also, the names of the crushing and compressive strengths should be checked.
  3. What was the water content of polyurethane concrete? It should be given in Table 2. The workability values should be explained in the paragraph under Table 3.
  4. Why the Authors presented both Fig 22 and Table 4. One of them should be removed. 
  5. In line 464. T should be written in lower case
  6. Title of section 5 (patents). Must be checked.

Reviewer 3 Report

Whilst this paper has some limited merits it would be restricted to a very limited set of readers whose bridges may fall under the Chinese style of the paper. The hypothesis is it is the mechanical degradation in the form of fatigue which causes the eventual breakdown of the deck slab, whereas in reality it is really much more complex due to thermal stresses, moisture causing ASR/DEF and corrosion.  Rarely would truck traffic alone be the cause of the breakdown.  That said the paper has some limited interest, particularly experiments and the remediation used.

Chinese writer must get away from using the improper use of the English usage of the the words "bearing capacity"  To a bridge engineer, bearings are the seats on a bridge, which have a load capacity or just bearing capacity.  This has nothing to do with the bridge deck.  What you strictly mean is load carrying capacity. But even that is fraught with ambiguity as there is a significant difference between the "safe applied live loads" and the "ultimate live load capacity." In the USA, this difference is a factor of 1.75 plus another 1.33 for dynamic impact [or 233%].  You must be clearer.  Also, simply do a search and replace of the words "bearing capacity" with ultimate [or safe if that's what you mean] live load capacity. 

Round 2

Reviewer 1 Report

Point 9: I meant that for clarity it would be good to put a scale in mm in figure 15 

Reviewer 3 Report

The revised version is better and is now OK to publish.  When changing from bearing capacity the word live can be removed as it is both redundant and implied, just say 'ultimate load capacity'  as this encompasses both dead, live and any other form of loading.

Author Response

This manuscript is a resubmission of an earlier submission. The following is a list of the peer review reports and author responses from that submission.

Round 1

Reviewer 1 Report

This paper is on fatigue, therefore the veracity of the claims and conclusions depends on fatigue results and their reproducibility.  Other than some photos, there is insufficient detail given on the experimental work in terms of its ability to be reproduced and thus independently reproduce the test results to verify the results.   Therefore, the claims are not solid. 

Also, there are too many references citing Chinese publications which are inaccessible in English.  The interpretation of the fatigue is non-traditional, and in my mind unbelievable.  So in my learned view the paper should not be published in its present form. 

Reviewer 2 Report

The Article Fatigue Performance of Rib Beam Bridge Slabs Reinforced with Polyurethane Concrete Based on the Damage Theory presents methodology, results/discussion in a clear way, as well as its contribution to the area. I have some suggestions: The title suggests a basically theoretical analysis. However, the experiments that support the theory are robust and deserve to be emphasized. In the Introduction, it would be interesting to inform the types of fatigue tests and the most used simulation models. The state of the art needs to be improved as there are recent studies. The use of Polyurethane Concrete Based must be justified by theoretical reference. Fatigue tests tend to present a large dispersion of results when repeated: this should be highlighted and justified (which shows the importance of the simulation results) as only one beam was tested for each situation. The results describe the graphs but could analyze the fiber action mechanism. I suggest that the conclusion 1 be more concise and direct.

Reviewer 3 Report

The reviewer thanks the authors for the numerical and experimental work regarding the problem of fatigue in prestressed concrete girders. The findings from this study could be useful information for field applications. In the reviewer’s opinion, the goal of the work must be better explained within Abstract, Introduction and Conclusions. Moreover, the publication in the “Materials, MDPI” is not recommended unless the following suggestions are taken into account within the article:

1) The current state of knowledge relating to the topic has not been covered and clearly presented, and the authors’ contributions are not emphasized. In this regard, the authors should make their effort to address these issues, by adding additional comments on the state of the art and the proposed aspects.

2) Objectives and information should be presented more clearly. Furthermore, additional comments should be added in regard to the practical value of this work, and how the industry can profit from this article.

3) Compared with other materials and technologies, the superiority of the polyurethane concrete reinforcement layers, to prevent fatigue deformations in bridge-girders, is not clearly explained. Please, review the corresponding sections.

4) Fatigue can cause concrete cracking and corrosion of tendons along bridge-girders and, consequently, provoke significant prestress losses and deflections during their service life. Please, refer to the aforementioned issues and cite the following references in the Introduction:
-  Stiffness of corroded partially prestressed concrete T-beams under fatigue loading. Mag. Concr. Res. (2018) 1800187.
-  Prestress loss diagnostics in pre-tensioned concrete structures with corrosive cracking. J. Struct. Eng. 146 (3) (2020) 04020013.
-  Novel method for identifying residual prestress force in simply supported concrete girder-bridges. Advances in Structural Engineering 24 (14) (2021).
A wider presence of references can make the article more popular and profitable by the scientific society.

5) Figures 1, 2, 3 and 6 do not seem to represent the specimens tested in the laboratory. Please, modify.

6) The authors did not mention about the prestressing state of the specimens tested. How much prestressing loading has been applied to the specimens before fatigue testing ? How much prestressing loading has been measured in the specimens after fatigue testing ? Moreover, Section 2.1.1., which treats the descriptions of the specimens, does not report the characteristics of prestressing tendons. Please, specify.

7) Compression tests on the used concretes have not been mentioned. Did the concrete characteristics of the specimens obtain by compression tests ? Please, specify and report clearly the experimental values achieved.

8) Section 2.2.2. The section should furnish more information regarding the types of finite elements (FE) adopted with corresponding amount and mesh sizes.

9) Fig. 13. There is a systematic error between numerical and experimental displacements. The error is probably due to the vertical displacements at the supports of the specimens which, in turn, have not been measured and subtractive from the midspan deflections for comparison with FE modeling. Please, specify.

10) Figures 16, 17, 19 and 20 do not report the comparisons with FE modeling. Please, specify the reasons.

11) Please, provide the frequency (or the period) of the recording data by all the equipment and devices used.

12) Please, provide the technical characteristics of all the equipment and devices used. In specific, how many strain gauges and dial indicators for displacement measurements have been utilized ? Particularly, from Figure 18, the number of strain gauges is not clear.

13) It is also not clear at which parts of the specimens tested (after fatigue cracking) the pictures are referred. Please, improve and make clear such information.

14) The further research, related to the topic of the article, should be underlined in the conclusions.

15) I suggest to the authors to edit all the text of the article with the help of a native English speaker. Grammar, punctuation, spelling, verb usage, sentence structure, conciseness, readability and writing style must be improved.